# Laser-Assisted Drug Delivery: A Systematic Review of Safety and Adverse Events

**DOI:** 10.3390/pharmaceutics14122738

**Published:** 2022-12-07

**Authors:** William Hao Syuen Ng, Saxon D. Smith

**Affiliations:** 1Greenslopes Private Hospital, Brisbane, QLD 4120, Australia; 2ANU Medical School, ANU College of Health and Medicine, Australian National University, Canberra, ACT 2601, Australia

**Keywords:** lasers, laser-assisted, laser-facilitated, laser-assisted drug delivery, skin, dermal, dermatology, topical, cutaneous, human, human studies, clinical manifestations, clinical signs, symptoms, safety, adverse events

## Abstract

Laser-assisted drug delivery (LADD) is an increasingly studied and applied methodology for drug delivery. It has been used in a wide variety of clinical applications. Given the relatively low barrier to entry for clinicians as well as ongoing research in this area, the authors aimed to review outcomes relating to safety in laser-assisted drug delivery. A systematic review was conducted, with the databases PubMed, Medline and Embase searched in September 2022. Included articles were those that mentioned laser-assisted drug delivery in human subjects that also reported adverse effects or safety outcomes. There were no language-based exclusions. Conference abstracts and literature reviews were excluded. The results were then tabulated and categorized according to the application of LADD. In total, 501 articles were obtained. Following deduplication, screening, and full text review 70 articles of various study designs were included. Common findings were erythema, oedema, pain, and crusting following LADD. Several notably more severe adverse effects such as generalized urticaria, infection, scarring and dyspigmentation were noted. However, these events were varied depending on the clinical use of LADD. Relevant negatives were also noted whereby no studies reported life-threatening adverse effects. Limitations included limited details regarding the adverse effects within the full texts, lack of follow-up, and risk of bias. In conclusion, there were multiple adverse effects that clinicians should consider prior to carrying out LADD, where treatment goals and patient tolerability should be considered. Further evidence is needed to quantitatively determine these risks.

## 1. Introduction

Laser-assisted drug delivery (LADD) is an increasingly utilized method to enhance the topical delivery of drugs. The primary concept of laser-assisted drug delivery is to facilitate increased topical absorption, commonly through skin [1]. The commonly used procedure is the administration of laser on the area to be treated followed by the topical application of the therapeutic substance. This may be limited to a single application or continued as a treatment regimen.

In recent years, laser-assisted drug delivery has received significant attention in its use for clinical and academic purposes, with increasing application to human subjects [2]. Some institutions have already utilized LADD over a period of years with published findings regarding their experiences [3,4]. 

Laser-assisted drug delivery has a relatively low barrier of entry as it utilizes equipment that practitioners may already own, particularly in its application for its use with skin. Given the wide variety of hypothesized indications for laser-assisted drug delivery, and its relative accessibility, it has the potential to have widespread adoption. As with any novel intervention, the aspect of safety should be emphasized.

A commonly cited source [5] states that its conceptualization was first established in 1987. Since then, Laser-assisted drug delivery has been studied for its various applications in oncology, analgesia, anaesthesia, and various medical indications.

The skin which forms the integumentary system possesses the properties of some impenetrability to molecules and drugs [6,7]. Laser assisted drug delivery has been regarded as a mechanism to overcome this, for increased drug delivery through the topical route [8].

The procedure of laser-assisted drug delivery itself is the utilization of various types of lasers on targeted skin [9], followed by the application of the topical substance desired to be administered. In vivo and in vitro studies [1,10,11] have shown that lasers are able to physically disrupt the skin barrier in order to increase the permeability of it to such substances. 

Fractional lasers are commonly studied for use in conjunction with laser-assisted drug delivery. These lasers deliver laser beams in a fractionated method, resulting in numerous vertical channels termed as microthermal treatment zones. These would allow the passage of topical substances [12]. Additional laser parameters may be manipulated to achieve a desired effect, for instance the depth of penetration, in order to target delivery to the epidermis or dermis.

LADD for the skin has been utilized using a wide variety of topical preparations, ranging from cosmetic (facial rejuvenation, keloids), medical, and oncologic conditions [1,12]. Additionally, it has been used as a precursor to other form of therapy, notably photodynamic therapy, through its use prior to the application of a photosensitizing agent [13].

Its efficacy has been studied for certain use cases, with increasing evidence regarding its efficacy particularly in the management of keloids [14] and actinic keratoses [15]. However, further research in this area is still warranted. The systematic review by Truong et al. [14] yielded studies with varying treatment regimens involving LADD. They have suggested that further high-quality RCTs are needed. Additionally, there are still relatively few systematic reviews with meta-analyses regarding the efficacy of LADD. This is exacerbated by the heterogeneity in administration protocols for LADD.

Given that LADD has the potential to have widespread adoption, with a significantly large variety of use-cases as well as therapeutic agents, the authors believe that it would be useful to summarize published evidence regarding the safety of LADD. In particular, the clinical manifestation of harms associated with LADD. 

## 2. Methods

A systematic search was conducted in September 2022, with databases PubMed, Medline and Embase searched. Inclusion criteria were articles mentioning laser-assisted drug delivery in human subjects with mention of adverse reactions or other data related to safety. A combination of search terms relating to laser-assisted drug delivery, safety, and adverse effects were used (Appendix A). There were no restrictions to language of the articles. Following deduplication, two authors screened the resultant title and abstracts, then excluded articles where deemed appropriate, following which the full texts of the remainder of the articles were assessed using the full texts. Conference abstracts and articles through which full texts were inaccessible due to copyright limitations were excluded. Literature review articles were excluded to avoid duplication of cases. The study including this protocol was registered in OSF Registry (Open Science Framework Registry) [16].

Data was extracted to identify the number of patients receiving laser-assisted drug delivery and reported adverse events or safety issues. Study characteristics such as study design and pharmaceutic agents used were also included. Relevant negative findings were also included.

## 3. Results

Studies were of various designs were captured within our search, ranging from single-patient case reports to larger trials; controlled and uncontrolled. There were a several retrospective reviews as well. A number of trials utilized intra-patient comparisons were noted as well, particularly in the field of studying keloid scarring, actinic keratoses, and melasma.

Most studies reported relatively similar adverse effects. The most commonly mentioned adverse effects included pain, erythema, crusting, and swelling. Notably severe adverse effects documented were worsening of pigmentary disorders or new pigmentary issues in the management of pigmentary disorders, scarring and infection. There was varying availability of the number or proportion of participants experiencing adverse effects. This information was included if it was reported.

A total of 501 articles were obtained from the initial search. The process for study selection was summarised in Figure 1 based on the PRISMA flow diagram [17]. Following deduplication, the title and abstracts for 314 articles were screened. Subsequently the full texts of 83 articles were sought for. Two studies were noted to be potentially meet inclusion criteria however these were excluded due to inaccessibility due to copyright restrictions [18,19]. After exclusion of articles that did not meet the inclusion criteria, 70 articles were included to be tabulated for the final analysis. For this, the results were divided into several groups with similar application of LADD.

The following tables and corresponding categories were used:Table 1: Skin cancers/oncology/pre-cancerous lesionsTable 2: ScarsTable 3: Pigmentary disordersTable 4: Hair loss/alopeciaTable 5: Acne vulgarisTable 6: AnalgesiaTable 7: OnychomycosisTable 8: Miscellaneous medicalTable 9: CosmeticTable 10: Non-therapeutic experiment

There are two main uses of laser-assisted drug delivery in this category-for the delivery of topical chemotherapeutics, and topical photosensitizers prior to photodynamic therapy. In both these conditions, there were relatively more side effects documented compared to other use cases for LADD in our study. The side effects captured included more severe effects less seen in the other applications of LADD such as scarring [21,23], purpura [33], hyperpigmentation [21,23,26,32,34], hypopigmentation [21,23] and bullae [26,32,34]. These adverse effects have been reported in LADD studies involving PDT and those without. Additionally, in the study by Lonsdorf 2022 [20], it was noted that 2 participants requested early termination of PDT following LADD due to pain. This could potentially reflect the relatively severe pain associated with LADD and PDT.

Another factor of interest is that all but one study quantified the proportion of participants that experienced adverse effects. This could reflect an increased vigilance of the authors in this area of study regarding the adverse effects of LADD.

The adverse effects regarding intralesional (injected) therapies for scarring was noted in Manuskiatti 2021 [36], Abd El-Dayem 2020 [37], Wang 2020 [40]–telangiectasia, hypopigmentation, skin atrophy. Multiple studies have investigated LADD for various types of scarring, with some directly comparing intralesional therapies as in Abd El-Dayem 2020 [37], Sabry 2020 [39] Park 2015 [43]. Various agents were used with LADD for scars with steroids being the most commonly used agent. Other agents included vitamin-C, growth factors, botulinum toxin and PRP. There were still several side effects that are usually associated with intralesional steroids which were present in LADD of steroids including telangiectasia mentioned in Wang 2020 [40] and dermal atrophy and telangiectasia in Waibel 2019 [41]. Future studies may be able to quantitatively compare this risk. Additionally, there were also mentions of hyperpigmentation following LADD for keloids/hypertrophic scars which could potentially represent post-inflammatory hyperpigmentation in this potentially predisposed patient cohort. However, nil systemic side effects were reported in these studies in those that reported the relative negative findings.

A significant number of studies have used laser-assisted drug delivery for the management of pigmentary conditions. Some of these studies recruited from populations that did not respond to topical therapy alone. In terms of the studies captured that used LADD in these conditions, it was noted that there was a notable number of pigment-related adverse effects noted such as the worsening of melasma in several patients as documented in Botsali 2022 [45] and Wanitphakdeedecha 2020 [49]. Additionally, hyperpigmentation in vitiligo was noted in Doghaim 2019 [50] and Huang 2019 [3].

Similar to the use of LADD for hypertrophic/keloid scarring, LADD has been investigated as a potential alternative to intralesional injections for alopecia areata and hair loss. The side effects reported were telangiectasia, pruritus, erythema, dandruff, and pain in several patients. All studies reported nil significant/serious adverse effects. Pain severity seems to be limited, with Hanthavichai 2021 [53] reporting tolerable pain and Bertin 2018 [56] reporting only mild pain. This could potentially be an advantage compared to intralesional injections, especially if multiple treatment sessions are needed. A common site for these conditions–the scalp–would also be comparatively more sensitive to pain. Hence, clinicians and patients may prefer LADD compared to intralesional injections if it is associated with less pain. Further studies may be done to quantify this. One limitation however, is the relatively limited number of patients studied for this application of LADD, with only 5 studies captured with the largest cohort of patients being 30 in Soror 2021 [54].

Acne vulgaris was another condition whereby LADD was utilized. Several side effects that were also seen in other therapeutic uses for LADD were observed such as erythema, desquamation, hyperpigmentation, oedema were noted. In Hædersdal 2008 [59] it was noted that 12/15 patients receiving LADD had pustular eruptions. One patient was managed as per cutaneous infection as well. The underlying condition of acne vulgaris may also be associated with this as well as pustular eruptions reported–it was not specified if the pustular eruptions were of similar characteristics or not to the patient’s usual acne. It was also not specified if this represents a worsening of the patient’s existing acne following the use of LADD.

A small number of studies have investigated the use of topical analgesia following application with laser. In these studies, a relatively small area (dorsum, deltoid) had laser application with a portable device prior to the application of a topical preparation of local anaesthetic. Pain was the main adverse effect following these trials however these studies all included the use of penetration of skin with needles/cannulation. It is noted that these articles were published between 2003 and 2006, with two studies with the same first author [60,61]. Further studies with larger sample sizes may be able to further inform outcomes related to safety as well as efficacy, should this application of LADD be utilized more frequently in the future.

Irradiation of the nails were used in conjunction with topical delivery of anti-fungal agents. These studies used laser application followed by a regime of topical application of antifungal agents. Adverse effects were limited to mainly pain, and some pinpoint bleeding, with pain reported in all studies. However, relative negative findings reported were notable for a relatively mild side effect profile. Bhatta 2016 [65] and Rajbanshi 2020 [66] reported nil contact dermatitis and dermatitis, respectively. Promisingly, this could be interpreted as the treatment regimen not causing significant surrounding irritation in those instances.

There were several studies in which LADD was used to deliver a variety of therapeutic agents, in use cases including medical and oncologic conditions. Bauer 2021 [70] was the only study within our review that investigated the delivery of a biologic therapeutic for chronic plaque psoriasis, with itching, redness, pain, and ulceration reported. There were also a few notably severe systemic effects captured such as GIT bleeding, abdominal cramping, hypertension, etc. which were explained by the authors to be unrelated to LADD. These systemic effects may be an area of which to be monitored in future studies of LADD involving biologic agents. 

Within the other studies in this category there were a few generic side effects seen in other LADD applications. There were relatively few adverse effects reported for the use of LADD in the use of conditions for palmar hyperhidrosis, macular amyloidosis, cicatricial ectropion and common warts.

There were 2 studies investigating LADD for the delivery of timolol for infantile hemangiomas. These studies were more rigorous than others in monitoring patient parameters including clinical and laboratory findings, with only local reactions reported.

One study also investigated the use of LADD for patch testing, including patch testing for patients with known allergies to certain substances with patches that contain said substances, which was implied to be an intended effect of the use of LADD. The side effects were limited for this study, however, this was the only study with this application for LADD.

There were two articles which reported the use of lasers with imiquimod for tattoo removal. Urticaria was mentioned as an adverse effect, however it was noted that one subject had recurrent and generalized urticaria and facial angioedema in Ricotti 2007 [84]. There were also localized changes that were noted with this use-case of LADD.

Hyaluronic acid, a combination of vitamin C and growth factors, and amniotic membrane stem cells were used for cosmetic purposes with LADD [80,81,82]. The side effects of which included generic effects with ablative lasers. It was noted that acneiform eruption was noted in a patient with amniotic membrane stem cell [82], which could represent infection.

Two experimental studies were included. Banzhaf 2016 [85] aimed to study the penetration of substances, and have selected fluorescein to be studied. The side effects included generic ones associated with AFXL. Oni 2013 [86] investigated the serum levels of serum lidocaine with the application of a combination topical anaesthetic cream after two different types of lasers. Serum levels of lidocaine did rise significantly however did not reach a level specified to be toxic specified by the authors. These patients had laser irradiance of the face only. There is a potential risk if larger areas were utilized that a dangerous level of local anaesthetic may be present systemically, however this was not able to be quantitatively ascertained in this study.

## 4. Discussion

The authors believe that to date, this is the first systematic review regarding the aspect of safety and adverse effects associated with laser-assisted drug delivery. The utilization of the search string was intended to capture studies that reported regarding the safety aspect of laser-assisted drug delivery, even when it was not to be the primary outcome of the study. Quantitative analysis of the results was not the main objective of our study, rather we intended to portray broadly the themes regarding safety in LADD. The results reflect studies of a variety of structures with significantly varying reporting regarding safety. Moving forward, with the increasing evidence to guide optimal protocols, the aspect of safety should also be considered alongside efficacy.

Limitations to our study include the limited description and characterization of adverse effects by certain authors in the full text, where additional information would have been valuable for analysis. Many studies did not clearly specify the number of participants that experienced adverse effects as well, only stating symptoms observed. Additionally, the limited number of studies that performed long-term follow up is another factor whereby the assessment of adverse effects that may last for a longer duration of time were not captured. Bias of the authors would be another factor as many studies were non-controlled for, and obtaining data was done in an opportunistic method, for instance in retrospective reviews. Apart from notable exceptions such as Oni 2013 [87], and Ma 2014 [79], that monitored for physiologic parameters and regular blood tests post-treatment, most studies relied on patient and clinician findings for adverse effects, rather than objective markers. In terms of limitation of our study design, the use of topically delivered medications following laser therapy would technically fit the criteria of laser-assisted drug delivery. However, this may not be reported as laser-assisted drug delivery. Hence, our review may not capture all published instances of medication application following laser irradiation.

A particular challenge as well in obtaining the results included the heterogeneity of language used to describe adverse effects–this was manifested by many papers stating that there were nil systemic adverse effects, or others stating that there were no severe adverse effects which would be open to interpretation by the reader, although it likely implies that there was no systemic adverse effects, disfigurement, or life-threatening complications. For purposes of readability, the descriptions used by authors were used verbatim, if possible, with contractions for readability and formatting applied where needed.

To be considered as well would be that the adverse effects of LADD are inevitably linked to adverse effects from the use of lasers themselves. The use of lasers for dermatologic purposes has its own risk of adverse effects. The side effects of pain, erythema, crusting, and oedema have all been previously documented adverse effects of the use of lasers [87,88]. More severe side effects such as hypopigmentation and hyperpigmentation have also been associated with the use of lasers alone. Various operation settings for lasers were used in each study captured. It would be likely that laser choice and laser power settings affect the occurrence of adverse effects as mentioned above.

It was noted that laser-assisted drug delivery has also been used on conjunction with photodynamic therapy [89]. Our findings show that there were multiple adverse effects as described above. The use of photodynamic therapy itself is associated with adverse effects [89,90,91]. From the studies investigating LADD with photodynamic therapy, there were similar side effects as with photodynamic therapy itself, including pain, erythema, discomfort and skin changes. Comparatively, these appear to have a higher rate of reported side effects compared to other uses of LADD however factors affecting this could be the mechanism of PDT itself, increased vigilance from the investigators and patient bias.

Of the more severe adverse effects were documented, one of which was generalized urticaria in a patient noted by Ricotti 2007 [84]. Localized urticaria has been reported as an adverse effect following laser therapy [92,93] however the authors have stated that this patient has had a generalized urticarial reaction. Allergic reactions similar to this have been previously reported with laser tattoo removal [94], in an immediate [95] or delayed presentation [96]. This has noted to be a rare complication of laser tattoo removal [94] however in the context of LADD it is important to consider that imiquimod may have had a role in this reaction. 

The aggravation of pigmentary conditions is also a significant concern. LADD has been used for melasma and vitiligo which are conditions under this category [97,98]. Worsening of the aforementioned concerns were mentioned in Li 2022 [46], Wanitphakdeedecha 2020 [49], and Botsali 2022 [45]. Additionally, hyperpigmentation in vitiligo was noted in Doghaim 2019 [50] and Huang 2019 [3], which are new cosmetic concerns. Prior studies have indicated that the use of lasers in these conditions should be done with caution [99]. These adverse changes may cause distress to patients hence the risks of which should be considered.

One of the adverse effects described with the use of lasers in dermatology is “downtime” with different types of lasers yielding different results [100]. There were a significant number of mentions of erythema in many studies used in LADD. This could be more important to certain patient cohorts compared to others. Certain substances and therapies used with LADD is to be used for oncologic purposes such as with 5-fluorouracil and PDT have already been significantly associated with erythema when used alone [101,102]. However, in the use of LADD for cosmetic or aesthetic purposes especially when an alternative, non-LADD treatment route is available, downtime should be considered. This is an area which is subjective however clinicians may wish to consider this aspect when considering LADD.

The relevant negative findings should also be emphasized. Many all articles have reported no systemic side effects or adverse reactions. Besides this, there were no life-threatening adverse effects such as anaphylaxis or systemic toxicity from topical absorption.

## 5. Conclusions

Most laser-assisted drug delivery side effects have been limited to local reactions similar to that of laser therapy, in line with those associated with lasers themselves. The process is generally well tolerated with some exceptions. Some severe and systemic side effects were noted such as dyspigmentation, scarring and more rarely urticaria. Variable safety outcomes have been reported with different use-cases of LADD. In vivo studies may be of use to further characterize risk.

## Figures and Tables

**Figure 1 pharmaceutics-14-02738-f001:**
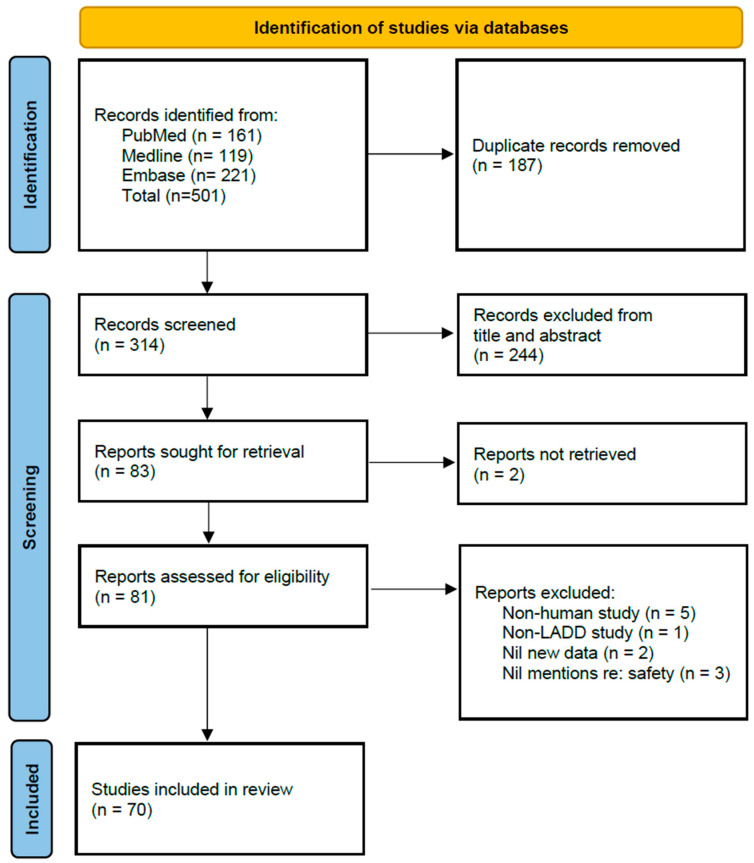
PRISMA-style flow diagram for study selection.

**Table 1 pharmaceutics-14-02738-t001:** Skin cancers/oncology/pre-cancerous lesions.

Author	Patient Cohorts	Number of LADD Recipients	Condition	Intervention	Laser	Adverse Events	Relevant Negative Findings
Lonsdorf 2022 [20]	Intrapatient comparison	18	AKs in organ transplant recipients	LADD + MAL-PDT vs. PDT	AFXL Er:YAG	Pain necessitating early termination of PDT (n = 2)	No therapy-associated systemic side-effects on day of rx and in F/U period
Fredman 2022 [21]	Single arm (follow up from previous study)	20	BCC (superficial & low-risk nodular)	LADD + Cisplatin + 5-FU	AFXL CO_2_	Persistent erythema (n = 5, <6 months; n = 2, @12 months), Hypopigmentation (n = 11 @12 months), Hyperpigmentation (n = 10 @3 months; n = 4 @6months) Scarring (n = 8 @6 months; n = 4 @12 months) Temporary crusting, inflammation	Nil severe adverse events
Paasch 2020 [22]	Prospective uncontrolled	46	Field cancerization (AK)	LADD + ALA-PDT (indoor light)	AFXL CO_2_	Pain (severe)	-
Wenande 2020 [23]	Single arm, prospective	19	BCC	LADD + 5-FU and cisplatin	AFXL CO_2_	Ooze (100%), Persistent erythema (83%), Hyperpigmentation (56%), Scarring (50%), Oedema, pruritus (32%), Mild hypopigmentation (17%)	Nil reported infection, systemic symptoms Nil cisplatin, 5-FU detected in blood 24h post-treatment (6 patients tested)
Pires 2019 [24]	Intrapatient comparison (split-arm)	15	AK	LADD + Acoustic pressure wave ultrasound + MAL-PDT vs. MAL-PDT	AFXL CO_2_	Mild pain (all, <3 h), Erythema, edema, crusts (all, <15 days)	Nil scarring on follow-up
Dairi 2018 [25]	Case series	4	Mycosis fungoides	LADD+MAL-PDT	AFXL CO_2_	Varying degrees of pain, local irritation, post-inflammatory hyperpigmentation lasting months (all)	Nil other adverse events noted
Kim 2018 [26]	Prospective uncontrolled	30	Lower extremity Bowen disease	LADD + MAL-PDT	AFXL Er:YAG	Erythema (n = 28 <7 days), Crusting (n = 24), Hyperpigmentation (n = 23), Burning sensation (n = 22), Pruritus (n = 21) Oedema (n = 9), Bullae (n = 3), Pain	-
Vrani 2018 [27]	Intrapatient comparison	50	AK	LADD + MAL-PDT vs. PDT	AFXL CO_2_	Erythema, oedema (all) Pustular eruption and crusting formation (n = 15).	Nil pain with AFXL, nil post-procedure scarring/pigmentary changes
Hsu 2016 [28]	Prospective single arm	28	Primary SCCis and sBCC	LADD + 5-FU	AFXL CO_2_	-	Nil treatment-related adverse events
Nisticò 2016 [29]	Intrapatient comparison	13	AK	LADD + Ingenol Mebutate vs. Ingenol Mebutate	AFXL CO_2_	Erythema (n = 13), Vesicles (n = 8), Oozing & crusts (n = 4)	
Braun 2015 [30]	Case report	1	Multiple AK	LADD + Ingenol mebutate	AFXL Er:YAG	-	No systemic side effects or safety concerns
Choi 2015 [31]	Prospective, dual arm	14	Actinic chelitis	LADD + MAL-PDT vs. MAL-PDT	AFXL Er:YAG	Mild-moderate pain (all, <7 d) Erythema (n = 13), Burning sensation (n = 13), Swelling, (n = 5) Haemorrhagic crusting (n = 3), Blistering (n = 2)	Nil systemic adverse effects
Choi 2015 [32]	Prospective, three-arm	64	AK	LADD + 2h MAL-PDT vs. LADD + 3h MAL-PDT vs. MAL-PDT	AFXL Er:YAG	2h vs. 3h Crust (86.8% vs. 82.8%), Erythema (78.1% vs. 80%), Hyperpigmentation (75.5% vs. 75.9%), Burning sensation (73.5%, vs. 75.9%), Pruritus (45.7%, vs. 52.4%) Oedema (7.9% vs. 6.9%), Bullae (4.6% vs. 6.2%)	Phototoxic adverse events were mild-moderate, short duration, did not require additional therapy
Helsing 2013 [33]	Intrapatient comparison	10	AK and wart-like lesions in organ transplant recipients	LADD + MAL-PDT vs. AFXL laser only	AFXL CO_2_	Intense inflammation and purpura (n=3) Erythema, oedema, pain (n=3)	-
Ko 2013 [34]	Prospective, dual arm	23	Facial AK	LADD + MAL-PDT vs. MAL-PDT	AFXL Er:YAG	Erythema (100%), Hyperpigmentation (100%, <20weeks)) Crust (100%), Burning sensation (73.8%), Pruritus (53.3%), Bleeding (31.8%), Scale (31.8%), Oedema (8.4%), Bullae (6.5%)	-

**Table 2 pharmaceutics-14-02738-t002:** Scars.

Author	Patient Cohorts	Number of LADD Recipients	Condition	Intervention	Laser	Adverse Events	Relevant Negative Findings
Machado 2021 [35]	RCT–2 arms	132	Scars (misc)	LADD + Vitamin C vs. LADD + Vitamin C and Growth Factors	AFXL ErYag	Nil reported	Nil local or systemic adverse reaction
Manuskiatti 2021 [36]	Intrapatient comparison (split-scar)	24	Hypertrophic scars	LADD + Clobetasol Propionate vs. LADD + Petrolatum	AFXL Er:YAG	Nil reported	Nil telangiectasias, dyspigmentation, skin atrophy, acneiform eruption
Abd El-Dayem 2020 [37]	Intrapatient comparison	30	Keloid scars	LADD betamethasone vs. intralesional triamcinolone acetonide	AFXL Er:YAG	Hyperpigmentation (n = 2)	No serious side effects
Neinaa 2020 [38]	Intrapatient comparison	45	Post-acne scars	LADD lypophylized-growth factors vs. LADD PRP	AFXL CO_2_	PRP vs. L-GFs Mild post-inflammatory hyperpigmentation (n = 8 vs. n = 13), Moderate post-inflammatory hyperpigmentation (n = 5 vs. n = 0) Acneiform eruption (n = 8) Downtime (2–5 days) Pain (<3 days)	No major side effects
Sabry 2020 [39]	Intrapatient comparison (split-scar)	20	Keloids/hypetrophic scars	LADD + BTX-A vs. Intralesional BTX-A	AFXL CO_2_	Pain, pruritus	-
Wang 2020 [40]	Prospective uncontrolled	41	Refractory keloids	LADD + Triamcinolone	AFXL CO_2_	Telengiectasia (n = 1), Hyperpigmentation (n = 4)	Nil allergic reaction, infection
Waibel 2019 [41]	Prospective–2 arms	20	Hypertrophic scars	LADD + Triamcinolone vs. LADD+5-FU	AFXL CO_2_	With LADD + triamcinolone-dermal atrophy, telangiectasia, persistent erythema	-
Kraeva 2017 [42]	Case report	1	Keloid scar (Fitzpatrick type VI skin)	LADD+ Triamcinolone acetonide	AFXL CO_2_	-	Nil complications or adverse events
Park 2015 [43]	Intrapatient comparison	10	Keloid scars (from BCG vaccination)	LADD + Desoxymethasone vs. AFXL+ Triamcinolone acetonide	AFXL Er:YAG	Micro-crust	No serious adverse reactions or events
Cavalié 2014 [44]	Retrospective cohort	23	Treatment resistant keloids	LADD betamethasone under occlusion	AFXL Er:YAG	Hypochromia in darker skin types, (n = 5 <1 month), Folliculitis (n = 3), Eczematous reaction to occlusive film (n = 3), Pain	-

**Table 3 pharmaceutics-14-02738-t003:** Pigmentary disorders.

Author	Patient Cohorts	Number of LADD Recipients	Condition	Intervention	Laser	Adverse Events	Relevant Negative Findings
Botsali 2022 [45]	Propsective trial	54	Melasma	LADD 5% Tranexamic Acid vs. LADD 5% Tranexamic + Oral Tranexamic Acid	AFXL Er:YAG	Increase in MASI (1.8–3.2 points) in skin types III, IV (n = 2)	No serious adverse effects
Li 2022 [46]	Intrapatient comparison	37	Melasma	LADD+ 10% Tranexamic acid vs. Laser + saline	755 nm picosecond alexandrite	Irritation (n = 17), Erythema (n = 36), Dryness (n = 36), Post-inflammatory hyperpigmentation (n = 10), Scaling (n = 3)	Nil hypopigmentation, infection, and scarring
Park 2021 [47]	Intrapatient comparison	25	Melasma	LADD + 3%Tranexamic acid, 5% Niacinamide, 1% Kojic acid vs. Nd-YAG alone	Q-switched Nd:YAG	Facial erythema (self-limiting)	Treatment was well-tolerated
Wang 2020 [48]	Prospective study	10	Melasma	LADD Tranexamic Acid	1927nm fractional thulium	Transient (skin) roughness, dryness, itching	All adverse effects were mild
Wanitphakdeedecha 2020 [49]	Intrapatient comparison	46	Facial melasma	LADD + 1.2% Tranexamic acid vs. Laser + Saline	FTL	Mild hyperpigmentation (n = 4) Mild pain	No scarring, hypopigmentation, or persistent erythema
Doghaim 2019 [50]	Intrapatient comparison	40	Stable vitiligo resistant to NBUVB	LADD + 5-FU + NBUVB vs. NBUVB	AFXL Er:YAG	Transient hyperpigmentation (all), Minimal scarring (n = 1), Tolerable pain (all)	Nil Koebnerization on follow-up
Huang 2019 [3]	Retrospective review; single arm	684	Stable vitiligo	LADD + Betamethasone	AFXL Er:YAG	Slight erythema, oedema (all) Hyperpigmentation (14.4%) Epidermal atrophy, telangiectasia, and hypertrichosis in lesions (0.14%)	Nil local infections, scarring, Koebner’s phenomenon, and aggravation of vitiligo
Badawi 2018 [51]	Intrapatient comparison (split-face)	32	Melasma	LADD + Hydroxychloroquine vs. hydroxychloroquine	AFXL Er:YAG	Erythema (all, <4 days), Superficial crusting, burning sensation (n = 7), Pruritus (n = 3), Superficial crusting	Nil worsening of melasma
Yan 2016 [52]	Intrapatient comparison	22	Non-segmental, resistant vitiligo	LADD + Betamethasone+ NBUVB vs. NBUVB	AFXL Er:YAG	Slight pain, burning sensation, edema, erythema (all), Micro-crust (~50%, <3 days)	Nil local infection, scarring, Koebner phenomenon, aggravation of vitiligo

**Table 4 pharmaceutics-14-02738-t004:** Hair loss/alopecia.

Author	Patient Cohorts	Number of LADD Recipients	Condition	Intervention	Laser	Adverse Events	Relevant Negative Findings
Hanthavichai 2021 [53]	Prospective trial	8	Androgenetic alopecia	LADD + PRP	AFXL CO_2_	Tolerable pain (n = 7, <several days) Mild pruritus (n = 2, <several days), Dandruff (n = 4, pts <2 weeks)	Analgesia not required before/after LADD Nil participants withdrew from study due to pain No serious adverse events such as infection, scarring, worsening of hair loss, and burn No scalp erythema and swelling were detected
Soror 2021 [54]	Intrapatient comparison	30	Alopecia areata	LADD + Triamcinolone vs. Intralesional Triamcinolone	AFXL CO_2_	Mild telangiectasia (n = 1)	No significant adverse effects
Majid 2019 [55]	Case series	10	Alopecia areata	LADD + Triamcinolone	AFXL CO_2_	-	No significant adverse effects, skin atrophy
Bertin 2018 [56]	Case series	5	Female/male pattern hair loss	LADD of topical finasteride, growth factors (vEGF, FGF, IGF, cooper peptide)	Non-ablative 1550 nm fractional Er:Glass	Mild pain, post-procedure transient erythema; (n = 2, <2 h)	“No significant side effects”

**Table 5 pharmaceutics-14-02738-t005:** Acne vulgaris.

Author	Patient Cohorts	Number of LADD Recipients	Condition	Intervention	Laser	Adverse Events	Relevant Negative Findings
Kim 2017 [57]	Prospective trial	14	Severe acne vulgaris	LADD + MAL + DL-PDT vs. MAL + DL-PDT	Non-ablative Er:glass	Erythema (n = 2, <1 week), Hyperpigmentation (n = 1), Tanning (n = 1) Pain	Nil bullae, crust, post-inflammatory hyperpigmentation
Jung 2011 [58]	Intrapatient comparison	22	Acne vulgaris	LADD + carbon lotion vs. Laser only. Noted that laser was after lotion application	Quasi-long and Q-switched ND:YAG	Transient erythema (all, <3 h), Mild dryness, mild desquamation (n = 15) Mild pain	Nil severe adverse events
Hædersdal 2008 [59]	Intrapatient comparison (split-face)	15	Acne vulgaris	LADD + MAL vs. Laser only	Long-pulsed dye laser	Erythema, oedema (n = 15) Pustular eruptions (n = 12) Yellow crusting mx with topical abx (n = 1) Moderate-severe pain	Nil long-term adverse reactions such as pigment changes, scarring

**Table 6 pharmaceutics-14-02738-t006:** Analgesia.

Author	Patient Cohorts	Number of LADD Recipients	Condition	Intervention	Laser	Adverse Events	Relevant Negative Findings
Singer 2006 [60]	RCT	30	Pre- cannulation analgesia	LADD + Lidocaine vs. lidocaine	AFXL Er:YAG	Mild pain with laser	Nil persistent erythema, infections
Singer 2005 [61]	RCT (Intrapatient comparison)	30	Pre- cannulation analgesia	LADD + Lidocaine vs. lidocaine	Er:YAG	Mild pain	Nil persistent erythema, infection, or scarring
Baron 2003 [62]	2x trials, Prospective dual arm	320	Needlestick (investigating analgesia efficacy)	LADD + Lidocaine vs. Laser + Placebo AND LADD + Lidocaine vs. Topical Lidocaine	Er:YAG	Mild pain, Erythema (n = 10)	-

**Table 7 pharmaceutics-14-02738-t007:** Onychomycosis.

Author	Patient Cohorts	Number of LADD Recipients	Condition	Intervention	Laser	Adverse Events	Relevant Negative Findings
Abdallah 2022 [63]	Intrapatient comparison (foot vs. foot)	21	Onychomycosis	LADD + PDT vs. PDT	AFXL CO_2_	Significant pain (n = 21), Pinpoint bleeding (n = 3)	All side effects were tolerated and temporary.
Koren 2017 [64]	Intrapatient comparison	60	Toenail onychomycosis	LADD + ALA-PDT vs. LADD + Amorolfine	AFXL CO_2_	Pain; scores 2.1–8.5/10	-
Bhatta 2016 [65]	Prospective, nil comparison	75	Onychomycosis	LADD + 1% terbinafine	AFXL CO_2_	Pain; mean 1.93/10	Nil bleeding, oozing, bacterial infection, contact dermatitis
Rajbanshi 2020 [66]	Prospective, dual arm	80	Onychomycosis	LADD + Terbinafine vs. Terbinafine	AFXL CO_2_	Pain (mean = 3.5/10)	Nil bleeding, infection Nil dermatitis, oozing. Nil observed medication cross-reaction

**Table 8 pharmaceutics-14-02738-t008:** Miscellaneous medical.

Author	Patient Cohorts	Number of LADD Recipients	Condition	Intervention	Laser	Adverse Events	Relevant Negative Findings
Agamia 2022 [67]	Intrapatient comparison	30	Palmar hyperhidrosis	LADD + BTX-A vs. Intradermal BTX-A	AFXL CO_2_	Pain	-
Johnson 2022 [68]	Retrospective review	33	Misc medical & cosmetic	LADD + Serum (15% Vitamin C, 1% Vitamin E, and 0.5% Ferulic acid)	AFXL CO_2_	Erythema (n = 6), Erythema+tenderness (n = 1), Erythema+mild bumpiness (n = 1), Skin peeling (n = 1), Pain+bleeding (n = 1)	All side effects resolved
Wang 2022 [69]	Retrospective review	94	Misc medical & cosmetic	LADD poly-l-lactic acid	AFXL CO_2_	-	Nil documented adverse effects. Nil filler nodules, delayed wound healing, prolonged erythema, and abnormal scarring
Bauer 2021 [70]	Intrapatient comparison	8	Chronic plaque-type psoriasis	LADD + Etanercept vs. Etanercept vs. AFXL alone	AFXL Er:YAG	Itching, redness, pain, ulceration *	Nil clinically significant deviation in lab results (chemistry, haematology, lipid panels)
Elazim 2021 [71]	Intrapatient comparison	32	Nail psoriasis	LADD + 0.1% Tazarotene vs. 0.1% Tazarotene	AFXL CO_2_	Mild-moderate pain (all, transient), Bleeding (n = 2) Periungual erythema (n = 2)	-
Helmy 2021 [72]	Prospective trial	11	Plaque psoriasis	LADD + Cyclosporine vs. Clobetasol	AFXL CO_2_	Mild burning and stinging (n = 2, < 48 h)	-
Sun 2021 [73]	Single arm prospective	30	Infantile hemangioma	LADD + Timolol	AFXL Er Yag	Detectable timolol systemically, 1.580–14.718 pg/mL in 8/20 patients. Erythema, oedema, blisters	Nil bradycardia, hypotension, hypoglycaemia, liver and kidney dysfunction, dyspnoea, lethargy, sweating in all subjects. Nil pigmentation, hypopigmentation, scars
Junsuwan 2020 [74]	Intrapatient comparison (palm vs. palm)	3	Palmar hyperhidrosis	LADD + BTX-A vs. nil treatment	AFXL	Pain–scale: 4-6/10	Nil pigmentation, textural changes. Nil change in hand dexterity, strength.
Shehadeh 2020 [75]	Intrapatient comparison	22	Nail Psoriasis	LADD + Betamethasone-Calcipotriol gel	Proximal and lateral nail folds–595nm PDL, Nail–AFXL CO_2_	Pain, Participants withdrew due to pain (n = 3) Local irritation/pain (<hours) erythema, purpura (<days)	-
Sobhi 2018 [76]	Intrapatient comparison (split-lesion)	10	Macular amyloidosis	AFXL vs. LADD + Topical Corticosteroid vs. LADD + Topical Corticosteroid + Vitamin C	AFXL CO_2_	Post-inflammatory hyperpigmentation (n = 1)	-
Lee 2017 [4]	Retrospective cohort	6	Cicatricial ectropion	LADD 5-FU	Various AFXL	-	No adverse effects other than AFXL-related
Park 2016 [77]	Prospective single arm	11	Common warts (paediatric patients)	LADD + Imiquimod	AFXL Er:YAG	Transient scabbing (n = 10), Erythema (n = 6), Pruritus (n = 4), (All adverse effects resolving in several days)	Nil serious adverse events necessitating additional treatment
Ma 2014 [78]	Prospective single arm	9	Infantile hemangiomas	LADD + Timolol	AFXL CO_2_	Pinpoint bleeding, fluid exudation (<1 day), Erythema, oedema (2–3 days), Dot crusting (5–7 days)	Nil detectable plasma timolol post-procedure. Nil significant change in HR, BP, BSL
Veremis-Ley 2006 [79]	Intrapatient comparison	14	For patch-testing	LADD + Patch Tests vs. Laser only vs. Patch Tests only	AFXL Er:YAG	Pruritus to positive- reaction sites, Crust and transient skin darkening (skin types IV, V)/lightening in skin types II, III (<10 days), TEWL increase (<48 h)	Nil reported side effects by patients, pain at laser-treated sites

* Several adverse events were classified as unrelated to trial: influenza, contact dermatitis on the neck, gastrointestinal bleeding, abdominal cramps, headache, constipation, arterial hypertension, hyperlipidaemia, bleeding at laser application site, common cold.

**Table 9 pharmaceutics-14-02738-t009:** Cosmetic.

Author	Patient Cohorts	Number of LADD Recipients	Condition	Intervention	Laser	Adverse Events	Relevant Negative Findings
Benzaquen 2021 [80]	Intrapatient comparison	20	Heloderma stigmas	LADD + Hyaluronic acid vs. LADD Saline	AFXL	Erythema, oedema, crust	Nil granuloma formation at 8 months F/U
Machado 2020 [81]	RCT	149	Periorbital wrinkles	LADD + Vitamin C vs. LADD + Vitamin C+ Growth Factors	AFXL Er:YAG	Nil reported	Nil adverse reaction locally or systemically
Widianingsih 2019 [82]	Intrapatient comparison	9	Phoaging	LADD Amniotic Membrane Stem Cell vs. Laser + Saline	AFXL Er:YAG	Erythema (all, <2 weeks), Mild pain (n = 7), Acneiform eruption (n = 2), Crusting	Nil post-inflammatory hyperpigmentation
Elsaie 2009 [83]	Intrapatient comparison	3	Tattoo removal	LADD Imiquimod vs. Laser + Vehicle cream	Nd:YAG	Moderate pain Pruritus (n = 1)	Nil pruritus, pain, burning, scarring, ulceration, pigmentary alterations, or vascular changes
Ricotti 2007 [84]	Intrapatient comparison	20	Tattoo removal	LADD Imiquimod vs. Laser + Placebo Cream	Q-switched Nd:YAG, frequency-doubled Nd:YAG laser, Q-switched alexandrite	Pruritus (60%) Erythema (55%) Scale (40%) Burning (35%), Erosions (30%), Poor healing of biopsy site (20%) Urticaria (10%) 1 subject-recurrent and generalized urticarial reaction with facial angioedema	-

**Table 10 pharmaceutics-14-02738-t010:** Non-therapeutic experiment.

Author	Patient Cohorts	Number of LADD Recipients	Condition	Intervention	Laser	Adverse Events	Relevant Negative Findings
Banzhaf 2016 [85]	Intrapatient comparison	11	Healthy skin	AFXL + Fluorescein	AFXL CO_2_	Transient oedema, erythema, micro-crusting	-
Oni 2013 [86]	Prospective, dual arm	10	To study LADD anaesthetic safety	LADD (Full ablative Er:YAG) vs. LADD (AFXL CO2) Delivery of 20% benzocaine, 6% lidocaine, and 4% tetracaine cream	Er:YAG and AFXL CO_2_	Post-treatment hyperpigmentation (n = 1, <1 mo), rx w hydroxychloroquine cream) Pain	For all patients in all groups, serum lidocaine and MEGX did not reach toxic levels, maximum was 0.935 μg/mL.

## Data Availability

The search string provided in Appendix A was used to search the databases PubMed, Medline, Embase as stated. Data was extracted from the included articles as tabulated and cited.

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
