# Peer review of "Laser-Assisted Drug Delivery: A Systematic Review of Safety and Adverse Events"

_pharmaceutics, 2022, doi:10.3390/pharmaceutics14122738_

Round 1
Reviewer 1 Report
In this review, the authors focus on the multiple adverse effects for the process of application using LADD in clinic. I think this review can be published in the journal of Pharmaceutics after major revision.
My comments:
1. Writing must be improved. Few examples:
Abstract: “Laser-assisted drug delivery (LADD is an increasingly studied and applied methodology for drug delivery”----missing of ). “Inclusion criteria included articles that mentioned laser-assisted drug delivery in human subjects that also reported outcomes related to safety such as adverse effects.”----long, incoherent, hard-to-read sentence.
Page 1: “The commonly used method of which this is done is by the administration of a laser on the intended targeted surface followed by the topical application of the desired substance which may be limited to a single application or a regimen following laser therapy.” ----too long, incoherent, hard-to-read sentence.
There are numerous grammatical errors and inconsistent sentences throughout the manuscript.
2. Page 22: “1. 1. Wenande E, Anderson RR, Haedersdal M. Fundamentals of fractional laser-assisted drug delivery: An in-depth guide to experimental methodology and data interpretation. Adv Drug Deliv Rev. 2020;153:169-184. doi:10.1016/j.addr.2019.10.003https://pubmed.ncbi.nlm.nih.gov/35976634/ evidence based guidelines 2022” Repetition of reference number for all references.
3. The sections of Introduction, Background, and Rationale should merge into one part.
4. The Paragraph 1 of Results section should be replaced by more concise, qualitative, and informative sentences related to laser-assisted drug delivery, rather than the processing method.
5. The nomenclature section is suggested to put at the before of the main manuscript.
6. The author enlisted a plenty of applications of LADD and displayed in different Tables. But they did not discuss their scopes, advantages, disadvantages, and clinical effect. More importantly, no discussion was provided that will give a scientific understanding of the aspects of limitations and challenges of LADD.
7. “The authors believe that to date, this is the first systematic review examining the safety aspect of laser-assisted drug delivery” The sentence does not make any sense at all.
8. The authors described various side effects or adverse reactions of LADD. However, they did not clarify the effective strategy for inhibiting side effects during LADD application, which is more important.

Author Response
- Writing must be improved. Few examples:
Abstract: “Laser-assisted drug delivery (LADD is an increasingly studied and applied methodology for drug delivery”----missing of ). “Inclusion criteria included articles that mentioned laser-assisted drug delivery in human subjects that also reported outcomes related to safety such as adverse effects.”----long, incoherent, hard-to-read sentence.
Page 1: “The commonly used method of which this is done is by the administration of a laser on the intended targeted surface followed by the topical application of the desired substance which may be limited to a single application or a regimen following laser therapy.” ----too long, incoherent, hard-to-read sentence.
There are numerous grammatical errors and inconsistent sentences throughout the manuscript.
Sentences above addressed and manuscript edited for brevity of sentences where applicable.
- Page 22: “1. 1. Wenande E, Anderson RR, Haedersdal M. Fundamentals of fractional laser-assisted drug delivery: An in-depth guide to experimental methodology and data interpretation. Adv Drug Deliv Rev. 2020;153:169-184. doi:10.1016/j.addr.2019.10.003https://pubmed.ncbi.nlm.nih.gov/35976634/ evidence based guidelines 2022” Repetition of reference number for all references.
Issue addressed.
- The sections of Introduction, Background, and Rationale should merge into one part.
Done.
- The Paragraph 1 of Results section should be replaced by more concise, qualitative, and informative sentences related to laser-assisted drug delivery, rather than the processing method.
Done.
- The nomenclature section is suggested to put at the before of the main manuscript.
Done.
- The author enlisted a plenty of applications of LADD and displayed in different Tables. But they did not discuss their scopes, advantages, disadvantages, and clinical effect. More importantly, no discussion was provided that will give a scientific understanding of the aspects of limitations and challenges of LADD.
Study scopes, have been further explored in the latest version of the manuscript.
Potential advantages and disadvantages have been further explored in the latest version of the manuscript. The main objective of this study is to investigate the occurrences of adverse effects and issues relating to safety with laser-assisted drug delivery. Commenting on advantages and disadvantages would be based on correlating outcomes with evidence, which in the case of LADD, is challenging as there is a lack of high-quality evidence to guide real-world practice.
Limitations and challenges regarding LADD from the aspect of safety were discussed in the manuscript. Issues to be considered such as pigmentary changes, downtime, and pain were addressed in the discussion section.
- “The authors believe that to date, this is the first systematic review examining the safety aspect of laser-assisted drug delivery” The sentence does not make any sense at all.
Sentence edited for clarity.
- The authors described various side effects or adverse reactions of LADD. However, they did not clarify the effective strategy for inhibiting side effects during LADD application, which is more important.
Our review intends to systematically analyze published literature regarding negative outcomes relating to safety; offering our opinions regarding methods to mitigate adverse effects would be beyond the scope of this review.
Reviewer 2 Report
General considerations
The clinical relevance claimed by the authors is to review outcomes relating to safety and side effects in laser-assisted drug delivery. The systematic review was performed considering good standards for search and inclusion/exclusion criteria to reunite and resume the goal information.
The manuscript is clear and well-written. I suggest the inclusion of a graph/figure to illustrate and summarize the results (for example, associating the main side effects with the "diseases/treatments").
Abstract
Line 11: Add “)”
Line 15: Remove “were”
Line 23: Adjust this sentence to make it clearer: “However, these varied based on the clinical use of LADD.”
Introduction
Line 49: Use LADD instead “laser-assisted drug delivery”
· The introduction should be improved by containing more explanation/details about the LADD technique itself and the types of lasers more commonly used.
Background
Lines 55, 58, 63: Use LADD instead “laser-assisted drug delivery”
Line 60: “In vivo” and “In vivo” should be in italic.
Line 71: add “of” in this sentence: “….as a precursor” of “other form….”
Results
Line 99: Correct “tite” for “title”
The tables were not mentioned/referenced in the text.
· The data from tables 1, 4, 6, 7, and 10 should be more explored, such as what was performed for the other tables, mentioning the references throughout the sentence.
· Both tables 1 and 4 have the same subtitle “LADD for skin cancers/oncology/pre-cancerous lesions”. Please correct the table 4 information.
Discussion
Line 219: Please specify in this sentence that it concerns both “safety and side effects” since there are other reviews related to LADD.
Line 241: Please clarify this sentence: “…however significant variations in reporting this occurrence exist; hence this study may not capture all published instances of this phenomenon”
• The laser irradiance or the laser type used in the protocols could be directly related to some of the side effects, mainly for pain. Even though these parameters were not considered in the tables and there is no standardized dosimetry in the studies, these aspects should be included in the discussion.
References
The authors reported that “there are still relatively few systematic reviews with meta-analyses regarding the efficacy of LADD”. However, both references [14 and 15] mentioned in the manuscript introduction could be more explored as a systematic review related to the topic in the discussion.
Author Response
Thank you for the kind review of our project. We appreciate the time spent and constructive feedback provided to improve our study.
General considerations
The clinical relevance claimed by the authors is to review outcomes relating to safety and side effects in laser-assisted drug delivery. The systematic review was performed considering good standards for search and inclusion/exclusion criteria to reunite and resume the goal information.
The manuscript is clear and well-written. I suggest the inclusion of a graph/figure to illustrate and summarize the results (for example, associating the main side effects with the "diseases/treatments").
Abstract
Line 11: Add “)”
Line 15: Remove “were”
Line 23: Adjust this sentence to make it clearer: “However, these varied based on the clinical use of LADD.”
Done.
Introduction
Line 49: Use LADD instead “laser-assisted drug delivery”
- The introduction should be improved by containing more explanation/details about the LADD technique itself and the types of lasers more commonly used.
Edits made. Introduction was edited based on journal editor's comment and other reviewer to include background information, including LADD technique and lasers used.
Background
Lines 55, 58, 63: Use LADD instead “laser-assisted drug delivery”
Line 60: “In vivo” and “In vivo” should be in italic.
Line 71: add “of” in this sentence: “….as a precursor” of “other form….”
Edits made, noted missing word following precursor – have used the term “precursor to”.
Results
Line 99: Correct “tite” for “title”
The tables were not mentioned/referenced in the text.
- The data from tables 1, 4, 6, 7, and 10 should be more explored, such as what was performed for the other tables, mentioning the references throughout the sentence.
- Both tables 1 and 4 have the same subtitle “LADD for skin cancers/oncology/pre-cancerous lesions”. Please correct the table 4 information.
Table categories mentioned in text. Data explored further. Corrected error.
Discussion
Line 219: Please specify in this sentence that it concerns both “safety and side effects” since there are other reviews related to LADD.
Line 241: Please clarify this sentence: “…however significant variations in reporting this occurrence exist; hence this study may not capture all published instances of this phenomenon”
- The laser irradiance or the laser type used in the protocols could be directly related to some of the side effects, mainly for pain. Even though these parameters were not considered in the tables and there is no standardized dosimetry in the studies, these aspects should be included in the discussion.
Clairified first point. Sentence edited for clarity. Appreciate the valuable comment regarding dosimetry have included it for discussion.
References
The authors reported that “there are still relatively few systematic reviews with meta-analyses regarding the efficacy of LADD”. However, both references [14 and 15] mentioned in the manuscript introduction could be more explored as a systematic review related to the topic in the discussion.
Articles were further explored